# Spatio-Temporal Gait Parameters of Hospitalized Older Patients: Comparison of Fallers and Non-Fallers

**DOI:** 10.3390/ijerph20054563

**Published:** 2023-03-04

**Authors:** Emilie Bourgarel, Clémence Risser, Frederic Blanc, Thomas Vogel, Georges Kaltenbach, Maxence Meyer, Elise Schmitt

**Affiliations:** 1Department of Geriatrics, La Robertsau Geriatric Hospital, University Hospital of Strasbourg, 83 Rue Himmerich, 67000 Strasbourg, France; 2Department of Public Health, Methods in Clinical Research, University Hospitals of Strasbourg, 67000 Strasbourg, France; 3Mitochondria, Oxidative Stress and Muscular Protection Group (EA-3072), Faculty of Medicine, University of Strasbourg, 67000 Strasbourg, France

**Keywords:** gait disorders, fall, older people

## Abstract

Gait disorders are predisposing factors for falls. They are accessible to rehabilitation and can be analyzed using tools that collect spatio-temporal parameters of walking, such as the GAITRite^®^ mat. The objective of this retrospective study was to find differences between the spatio-temporal parameters in patients who had fallen compared to patients who did not fall in a population of older patients hospitalized in acute geriatrics department. Patients over 75 years were included. For each patient, spatio-temporal parameters were collected using the GAITRite^®^ mat. The patients were divided into two groups according to whether they had a history of fall. The spatio-temporal parameters were compared between the two groups and in relation to the general population. Sixty-seven patients, with an average age of 85.9 ± 6 years, were included. The patients had comorbidities, cognitive impairment and were polymedicated. The mean walking speed was 51.4 cm/s in non-fallers group and 47.3 cm/s in fallers group (*p* = 0.539), indicating pathological walking in comparison with the general population of the same age (average 100 cm/s). No association was found between the spatio-temporal parameters and fall, probably linked to many confounding factors such as the pathogenicity of walking of our patients and their comorbidities.

## 1. Introduction

Learning to walk takes place during early childhood and becomes automatic and natural in adulthood. As we age, it progressively become more complex and therefore require more attention [1]. The prevalence of gait disorders in older people is high and has been estimated at 35% by Verghese et al. in a population of adults aged 70 years and older living at home in the state of New York, United States [2]. These gait disorders can lead to falls, which represent a major public health issue in terms of hospitalization, morbidity and cost. One third of the subjects over 65 years old and half of the persons over 80 years old had fallen at least once per year [3], which represents 450,000 falls each year in France [4]. It therefore seems necessary to investigate the gait of older adults in order to improve it and thus try to reduce the risk of fall. Spatio-temporal gait parameters can be recorded using the GAITRite**^®^** mat [5], a computerized walkway (701 × 90 cm) with embedded pressure sensors, which is widely used in clinical and research settings, and for which excellent reliability has been reported in many studies [6,7].

Several studies analyzing the link between spatio-temporal parameters of walking and falls have already been performed, but most of them were carried out in a population of older people living at home and presenting no or few comorbidities [8,9,10,11]. Very few studies have been performed in a frail older population with multiple comorbidities or polymedication [12], which is the case for most patients hospitalized in acute geriatric department. The consequences of a fall in this frail population, in whom the risk factors for falls accumulate, can be dramatic, and the study of gait disorders is even more important.

The objective of our study was to evaluate the spatio-temporal gait parameters using the GAITRite**^®^** mat in patients hospitalized in acute geriatrics station and to compare them between a population of patients with an history of fall and a population of patients with no history of fall.

## 2. Materials and Methods

This study was a retrospective, observational, single-center study.

### 2.1. Study Population

Patients were retrospectively included if they were hospitalized in an acute geriatric department at the University Hospital of Strasbourg between June 2019 and February 2020, were older than 75 years of age and had autonomously walked on a GAITRite^®^ mat with or without technical walking aids during a physiotherapy session during their hospitalization. The study population consisted of two groups, a fallers group that consisted of patients with a history of fall, and a non-fallers group with patients with no previous history of fall. The study was approved by the local ethics committee and was conducted following the principles of the Declaration of Helsinki. All patients gave their consent to participate in this study. The study has been registered with ClinicalTrials.gov under the codename NCT05152069.

### 2.2. End-Points

The primary end-point of our study was to compare spatio-temporal gait parameters between the fallers group and the non-fallers group. The secondary end-points were to compare the spatio-temporal gait parameters between the study population and the general population, between single-task gait and dual-task gait, and to compare the demographic characteristics between the two groups of the study population.

### 2.3. Data Collection

Patients’ medical records were reviewed, and collected data included: demographic data (age, gender, institutionalized or at home); neurological, rheumatological, cardiovascular and psychiatric history, Charlson Comorbidity Index; clinical examination focusing on the presence of extra-pyramidal syndrome, undernutrition, hypoacusis, low visual acuity, cognitive disorders (with the value of the Mini-Mental State Examination (MMSE)) and presence of technical walking aid; medication history (number of drugs, type of drugs); reason for hospitalization; history of fall and presence of post-fall syndrome. Regarding the history of falls, the information was also asked orally to the patient.

The study population gait parameters were assessed using the GAITRite**^®^** mat. Before the beginning of a walk, the length of both legs of each patient was measured. The GAITRite**^®^** program requires there data to analyze the gait correctly and accurately. The walk had to start one meter before and end one meter after the mat to exclude the acceleration and deceleration period. Patients performed a round trip on the mat with their usual technical aid. Gait parameters were recorded in single-task during the first one-way and in dual-task during return. The dual-task consisted of counting down from fifty to zero, from one to one. Recorded GAITRite**^®^** data were extracted automatically from the software and consisted of gait speed, cadence, step time, cycle time, step length, stride length, heal–heal base support, single support, double support, swing, and stance. The general population normative values used for comparison were taken from a study conducted in Olmsted County, Minnesota, which provided baseline values for gait parameters in older adults (over 70 years) on the GAITRite mat (Appendix A) [13].

### 2.4. Data Analysis

In a first step, we compared demographic characteristics between fallers and non-fallers. Continuous variables were presented as means or medians and were compared using Wilcoxon test, categorical variables were presented as frequencies, and percentages and were compared using Chi2 tests of conformity or exact Fisher test. For the second step, we compared the gait parameters between the study population, stratified by age and gender, and the reference values of the general population. For each parameter, the mean difference between the study population and the reference value was calculated and then compared to 0 using a Student’s *t* test. To consider the multiplicity of tests and thus the increase in alpha risk, an adjustment of the *p* values was performed. For the third step, we compared the gait parameters between fallers and non-fallers, using Wilcoxon rank sum test (Mann–Whitney U test). The variability of gait parameters has been reported as a predictive factor for falls [14]; therefore, gait parameter variation coefficients (standard-deviation/mean × 100) were calculated. Additionally, these coefficients were compared between fallers and non-fallers with Students’ *t* test. In a final step, the gait parameters recorded during a single task were compared to dual-task recorded parameters. For each parameter, the mean difference between single-task and dual-task was calculated and compared to 0 using a paired Student’s *t* test, with a *p* value adjustment.

## 3. Results

### 3.1. Study Population

Among 67 patients in the study sample, 46 were categorized as fallers and 21 as non-fallers. At baseline, average age was 85.9 years, 60% were women and 93% lived at home. There were significantly more patients with a history of stroke (mainly ischemic) in the faller population compared with non-faller patients. The patients who fell had significantly more osteoporosis than the patients who did not fall. However, there was no statistically significant difference in the presence of traumatic fracture between fallers and non-fallers. On the cardiovascular side, the patients who fell were significantly more hypertensive than non-fallers. Of the 67 patients, 47 (70%) were cognitively impaired, with a mean MMSE of 19.4 +/− 6.4, and there was no statistically significant difference between the two groups. The mean Charlson score was 7.00 +/− 1.87, and there was no significant difference between the two groups. A large majority of patients had multiple medications with a mean of 7.04 +/− 3.25 medications at admission, with no significant difference between the two groups. Two-thirds of the study population used a technical aid to move around, mainly a Rollator or a cane/crutch. Most of the patients in the faller population were hospitalized following a fall, but there was no significant difference regarding the other reasons for hospitalization. The demographic characteristics of the study population are shown in Table 1.

### 3.2. Gait Parameters and Comparison with Reference Values

Gait parameters of the study population were compared to reference values, stratifying by age, gender and fallers or non-fallers group. The study population showed a statistically significant difference compared to the general population for many values. Walking speed, step and stride length were almost twice lower in the study population compared to the general population (about 100 cm/s), and the double support phase was twice longer in the study population. Gait parameters and comparison with reference values are shown in Table 2.

### 3.3. Gait Parameters and Comparison between Fallers and Non-Fallers

The mean walking speed was 51.4 ± 19.9 cm/s in the non-fallers group and 47.3 ± 17.7 cm/s in the fallers group. In both groups, the double support phase averaged >40% of the gait cycle, the step length was <40 cm, and the stride length was <70 cm. In both single and dual tasks, there was no statistically significant difference between fallers and non-fallers for any spatio-temporal gait parameters. Nine patients were unable to perform the dual-task walk. Gait parameters and comparison between fallers and non-fallers are shown in Table 3.

### 3.4. Variability of Gait Parameters between Fallers and Non-Fallers

Variation coefficients were used to reflect variability of gait parameters. In the study population, the variation coefficients ranged between 7.8% and 19.1%, which is significantly more than the variability of the general population (range between 3% and 8%) [13]. No significant difference in variability between fallers and non-fallers was observed, and the variability of parameters increased during the dual task. The variability of gait parameters between fallers and non-fallers is shown in Table 4.

### 3.5. Gait Parameters and Comparison between Single-Task and Dual-Task

Comparison between single-task and dual-task revealed a shorter step length, a longer double support phase, and a lower walking speed. These results are shown in Table 5.

## 4. Discussion

### 4.1. Study Population Characteristics

The analysis of the study population characteristics reveals a very high average age (86 years old), a large number of comorbidities (Charlson score of 7.0 +/− 1.87), the presence of cognitive disorders for 70% of the patients, polymedication (7.04 drugs on average) and the use of technical aids for two thirds of the population. Statistically significant differences between the population of fallers and non-fallers were also observed. Fallers have significantly more history of stroke, high blood pressure and osteoporosis than non-fallers, but no statistically significant difference in the presence of traumatic fracture.

Since patients who fall have significantly greater history of ischemic stroke and hypertension, we can hypothesize that they fall because of a greater proportion of brain vascular damage. Many studies report that a greater number of white matter hyperintensities are associated with a greater risk of falls [15,16,17]. However, one study by Srikanth et al. in 2009 showed that this would also be associated with an increase in the variability of the spatio-temporal gait parameters, which was not found in this study [15], probably due to the large number of comorbidities in both patients with and without falls that may also affect gait parameter. The presence of a more significant vascular history in fallers may also raise suspicion of increased executive impairment. Several studies have shown that executive function is related to functional status and that impaired executive function is associated with impaired walking and balance [18,19,20].

### 4.2. Gait Parameters of the Study Population

Analysis of spatio-temporal gait parameters compared with reference values of the general population revealed the pathogenicity of walking of the entire study population. The walking speed was slow (around 50 cm/s), the step and stride length was shortened (around 35 and 67 cm, respectively), and the double support phase was increased (>40%). The pathogenicity of the walking can be explained by the baseline characteristics of our population mentioned above. The influence of frailty criteria, comorbidities, and polymedication on spatiotemporal gait parameters was studied by Taller-Kall et al. in 2015 [12]. They were able to demonstrate that these four factors had a significant influence on the spatiotemporal gait parameters, with a notable slower walking speed and a shorter step length. In a second study by Härdi et al. [21], the significant impact of a walking aid on spatiotemporal parameters was also demonstrated.

Furthermore, in view of the pathological aspect of walking in patients who do not fall, it is likely that they will fall in the near future. We can therefore ask ourselves the question of the clinical relevance of identifying a faller and a non-faller in a population of patients hospitalized in the acute geriatrics department (frail, with comorbidities and polymedication).

### 4.3. Gait Parameters and Fall

In our study, in both single- and dual-task, no association was found between the spatio-temporal gait parameters and fall. In 2014, Mortaza et al. conducted a meta-analysis of 17 articles on the relationship between falls and spatio-temporal parameters of walking. They showed that the variability of the stance phase, gait speed, step length and stride length are the gait parameters that show the most difference between fallers and non-fallers. However, they pointed out that patient selection criteria and methodologies were inconsistent and that the majority of results are controversial. They concluded that spatio-temporal gait parameters are not sufficient enough to predict falls in elderly subjects. Furthermore, all the patients were elderly people in good health, with no notable neurological or rheumatological history, and they did not require any technical assistance for walking, unlike the patients in our study [22].

Few studies have been performed in a population of elderly, frail patients with comorbidities and polymedication. Verghese et al. conducted a prospective study of 597 adults in 2009, average age 80.5 years, and investigated the association between changes in spatio-temporal gait parameters and the occurrence of a fall adjusted for age, sex, education, falls, chronic illnesses, medications, cognition and disability. They found a mean gait speed of 92.8 +/− 24.1 cm/s, and a slower gait speed was associated with higher risk of falls in the fully adjusted models. In our study, the average speed in all our patients was 48.6 cm/s, which would reflect a major fall risk. Variability in spatiotemporal gait parameters, primarily variability in stride length, was also associated with fall risk [10]. This is the marker most frequently found in studies in association with the risk of falling. It is even used as a predictive factor for falls by some authors [14] and would be accessible to rehabilitation [10,23,24]. This difference tends to be significant in our study; however, our patient population was much smaller.

### 4.4. Strengths and Limitations

The main strength of our study was the use of the GAITRite**^®^** mat, which allowed us to have reliable, objective, comprehensive and precise data of the spatio-temporal gait parameters. Additionally, most of the patients had a gait parameter record during the dual-task, which allows for a more accurate and objective evaluation of gait disorders [25,26]. However, our study has limitations. Due to the retrospective design, missing data may have occurred for demographic characteristics, such as history of falls, which may limit internal validity. In addition, the small number of patients included may have decreased the power of our study, which may have accounted for the absence of statistically significant results. Furthermore, the monocentric design may limit the external validity of our study, as the study population may not be representative of all patients hospitalized in acute geriatrics.

## 5. Conclusions

In conclusion, in this population, no association was found between spatiotemporal gait parameters and fall. The absence of association may be linked to many confounding factors such as the pathogenicity of walking of the patients and their comorbidities. In a population of older adults with many comorbidities, polymedication, and cognitive impairment, there may be no association between gait parameters and fall at all, but studies with a larger number of patients are necessary to confirm this hypothesis.

## Figures and Tables

**Table 1 ijerph-20-04563-t001:** Demographic characteristics of the study population.

	Non-Fallers	Fallers	*p*-Value
	*n* = 21	*n* = 46	
Mean Age, years (s.d.)	85.9 (5.9)	85.9 (6.1)	0.802
Living at home, *n* (%)	19 (90)	43 (93)	0.65
Female, *n* (%)	13 (62)	27 (59)	0.8
Mean Charlson score (s.d.)	6.90 (2.02)	7.04 (1.81)	0.64
Neurological history, *n* (%)			
Ischemic stroke	2 (9.5)	15 (33)	**0.044 ***
Hemorrhagic stroke	0 (0)	4 (8.7)	0.3
Peripheral neuropathy	2 (9.5)	3 (6.5)	0.65
Parkinson’s syndrome	1 (4.8)	4 (8.7)	1
Extra pyramidal syndrome	6 (29)	8 (17)	0.34
Rheumatological history, *n* (%)			
Osteoarthritis	6 (29)	14 (30)	0.88
Osteoporosis	2 (9.5)	15 (33)	**0.044 ***
Traumatic fractures	3 (14)	17 (37)	0.06
Cardiovascular history, *n* (%)			
Persistent atrial fibrillation	7 (33)	17 (37)	0.77
Hypertension	12 (57)	37 (80)	**0.046 ***
Diabetes	4 (19)	9 (20)	1
Ischemic heart disease	2 (9.5)	9 (20)	0.48
Psychiatric history, *n* (%)			
Depressive syndrome	6 (29)	15 (33)	0.74
Chronic alcoholism	4 (19)	4 (8.7)	0.25
Physical examination, *n* (%)			
Undernutrition	4 (19)	7 (15)	0.73
Hypoacusis	6 (29)	6 (13)	0.17
Low visual acuity	3 (14)	5 (11)	0.7
Post-Fall syndrome	0 (0)	2 (4.3)	1
Cognitive variables			
Cognitive disorders, *n* (%)	14 (67)	33 (72)	0.67
Mean MMSE score (s.d.)	21.8 (4.7)	18.3 (6.8)	0.089
Medication at hospital admission			
*N* of drugs, mean (s.d.)	6.38 (3.17)	7.35 (3.27)	0.22
Antihypertensive drugs, *n* (%)	16 (76)	38 (83)	0.53
Anticoagulants, *n* (%)	5 (24)	16 (35)	0.37
Anti-platelet aggregation, *n* (%)	8 (38)	17 (37)	0.93
Antiarrhythmic drugs, *n* (%)	3 (14)	6 (13)	1
Anti-pain drugs, *n* (%)	11 (52)	32 (70)	0.17
Benzodiazepines, *n* (%)	11 (52)	22 (48)	0.73
Technical walking aid, *n* (%)	13 (62)	31 (67)	0.66
Walker	1 (4.8)	4 (8.7)	1
Rollator	6 (29)	11 (24)	0.68
Crutches/Canes	6 (29)	16 (35)	0.62
Reason for hospitalization, *n* (%)			
Fall	0 (0)	27 (57)	**<0.001 ***
Ischemic or Hemorrhagic stroke	0 (0)	3 (6)	0.546
Gait Disorders	1 (4)	3 (6)	1
Anorexia	3 (14)	3 (6)	0.368
Rheumatological reason	2 (9)	0 (0)	0.095
Cardiac decompensation	2 (9)	1 (2)	0.223
Gastroenterological reason	2 (9)	0 (0)	0.095
Behavioral disorders on cognitive disorders	4 (19)	1 (2)	0.031
Psychiatric reason	2 (9)	0 (0)	0.095
Urological reason	4 (19)	2 (4)	0.072
Pneumonia	1 (4)	5 (11)	0.0657
Cancer discovery	0 (0)	1 (2)	1
MMSE, Mini-Mental State Examination			

* Statistically significative.

**Table 2 ijerph-20-04563-t002:** Gait parameters and comparison with reference values.

Parameter	Gender	Non-Fallers		Fallers	
		Mean Difference *	*p* Value	Mean Difference *	*p* Value
Gait speed (cm/s)	Women	+54.18	**<0.001**	+52.71	**<0.001**
	Men	+46.96	**<0.001**	+59.69	**<0.001**
Cadence (step/min)	Women	+28.08	**0.002**	+25.48	**<0.001**
	Men	+5.64	0.138	+19.31	**0.005**
Step time (s)	Women	−0.27	0.084	−0.26	**0.017**
	Men	−0.03	0.138	−0.23	0.087
Step length (cm)	Women	+22.11	**<0.001**	+21.72	**<0.001**
	Men	+24.87	**<0.001**	+28.80	**<0.001**
Stride length (cm)	Women	+44.98	**<0.001**	+43.91	**<0.001**
	Men	+50.37	**<0.001**	+58.26	**<0.001**
Base of support (cm)	Women	−2.95	0.089	−4.08	**<0.001**
	Men	−4.65	0.082	−2.57	**0.036**
Swing (% cycle)	Women	−4.43	0.134	−6.46	**0.015**
	Men	−9.64	**<0.001**	−5.37	0.078
Stance (% cycle)	Women	−8.66	**0.006**	−7.31	**<0.001**
	Men	−5.23	**0.001**	−8.16	**<0.001**
Single support (% cycle)	Women	+8.79	**0.006**	+7.48	**<0.001**
	Men	+5.37	**0.001**	+8.37	**<0.001**
Double support (% cycle)	Women	−17.65	**0.006**	−15.63	**<0.001**
	Men	−10.53	**0.001**	−16.13	**<0.001**

* Mean difference = reference value − study population value.

**Table 3 ijerph-20-04563-t003:** Gait parameters and comparison between fallers and non-fallers.

Parameter	Task	Non-Fallers Mean (±SD)		Fallers Mean (±SD)		*p* Value	
Gait speed (cm/s)	Single	51.4 (±19.9)		47.3 (±17.7)		0.539	
	Dual	42.4 (±17.7)		41.9 (±16.2)		0.987	
Cadence (step/min)	Single	87.2 (±18.2)		83.3 (±20.7)		0.402	
	Dual	75.8 (±16.6)		74.4 (±21.2)		0.562	
		Left	Right	Left	Right	Left	Right
Step time (s)	Single	0.8 (±0.5)	0.7 (±0.2)	0.8 (±0.4)	0.9 (±0.5)	0.32	0.317
	Dual	0.9 (±0.4)	0.8 (±0.2)	0.9 (±0.4)	0.9 (±0.3)	0.608	0.788
Cycle time (s)	Single	1.5 (±0.6)	1.5 (±0.6)	1.6 (±0.8)	1.6 (±0.8)	0.41	0.406
	Dual	1.7 (±0.5)	1.7 (±0.5)	1.8 (±0.7)	1.8 (±0.7)	0.86	0.568
Step length (cm)	Single	33.6 (±11.2)	35.1 (±10.8)	33 (±8.7)	33.4 (±9.4)	0.792	0.429
	Dual	32.3 (±10.5)	33 (±11.3)	33.2 (±8)	33.3 (±9.5)	0.693	0.98
Stride length (cm)	Single	68.9 (±19.6)	69.1 (±19.7)	66.8 (±17.7)	66.8 (±17.5)	0.866	0.803
	Dual	65.7 (±20.4)	65.5 (±20.5)	66.8 (±17)	66.7 (±17)	0.794	0.807
Base of support (cm)	Single	12.8 (±4.9)	12.8 (±4.9)	12.8 (±4)	12.8 (±4)	0.941	0.941
	Dual	14.1 (±5.1)	14.1 (±5.1)	13.2 (±4.5)	13.2 (±4.6)	0.551	0.529
Single support (%)	Single	28.8 (±6.5)	27.7 (±6.3)	28.3 (±6.4)	27.6 (±6.9)	0.665	0.892
	Dual	27 (±7.3)	26.1 (±6.3)	27.2 (±7.5)	26.7 (±6.7)	0.893	0.608
Double support (%)	Single	43.7 (±12.1)	43.9 (±11.8)	44.5 (±13.6)	44.6 (±13.5)	0.914	0.829
	Dual	47.3 (±12.8)	47.5 (±13.1)	46.5 (±13.7)	46.4 (±13.4)	0.769	0.769
Swing (%)	Single	27.8 (±6.2)	28.8 (±6.5)	27.6 (±6.7)	28.3 (±6.5)	0.876	0.7
	Dual	26 (±6.4)	27 (±7.1)	26.8 (±6.7)	27.1 (±6.5)	0.632	0.827
Stance (%)	Single	72.3 (±6.2)	71.3 (±6.6)	72.4 (±6.7)	71.7 (±6.5)	0.855	0.685
	Dual	74 (±6.4)	73 (±7.1)	73.2 (±6.7)	72.9 (±6.5)	0.638	0.827

**Table 4 ijerph-20-04563-t004:** Variability of gait parameters between fallers and non-fallers.

Parameter	Task	Non-Fallers Variation Coeff. (%)		Fallers Variation Coeff. (%)		*p* Value	
		Left	Right	Left	Right	Left	Right
Step time	Single	7.9	11.1	14.08	14.82	0.06	0.45
	Dual	14.03	10.89	14.65	16.99	0.89	0.11
Step length	Single	12.33	13.86	13.78	14.46	0.63	0.87
	Dual	11.98	13.5	13.23	14.03	0.43	0.81
Stride length	Single	8.53	8.72	9.74	9.96	0.28	0.31
	Dual	8.86	8.92	9.98	10.04	0.25	0.27
Swing	Single	11	10.97	12.20	13.49	0.49	0.31
	Dual	16.73	11.56	14.27	13.83	0.63	0.2
Stance	Single	9.65	8.7	11.73	12.07	0.51	0.32
	Dual	10.03	10.67	13.54	14.75	0.27	0.24
Single support	Single	10.97	11	13.49	12.2	0.31	0.49
	Dual	11.56	16.73	13.83	14.27	0.20	0.63
Double support	Single	14.77	15.17	17.39	18.32	0.64	0.57
	Dual	14.27	15.21	18.98	19.14	0.22	0.33
Stride speed	Single	10.56	10.66	12.34	12.95	0.25	0.2
	Dual	11.81	11.65	14.63	14.05	0.15	0.17

**Table 5 ijerph-20-04563-t005:** Comparison of gait parameters between single-task and dual-task.

Parameters	Mean Difference *	*p* Value
Gait speed (cm/s)	−7.04	<0.001
Step length (cm)	−1.39	<0.001
Stride length (cm)	−2.44	0.003
Step time (s)	+0.09	<0.001
Swing time (s)	+0.03	<0.001
Stance time (s)	+0.15	<0.001
Single support time (s)	+0.03	<0.001
Double support time (s)	+0.12	<0.001

* Mean difference = dual-task value − single-task value.

## Data Availability

All data generated or analyzed during this study are included in this article. Further enquiries can be directed to the corresponding author.

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
