# Peer review of "Spatio-Temporal Gait Parameters of Hospitalized Older Patients: Comparison of Fallers and Non-Fallers"

_ijerph, 2023, doi:10.3390/ijerph20054563_

Round 1

Reviewer 1 Report

Thank you for the good research. It seems that the paragraph of the thesis needs to be corrected. Overall, the length of paragraphs is short, hindering understanding of the content. I think it will be a better thesis if it is modified according to the comments below.

Review Comments

Manuscript ID : IJERPH-2218917

Title : Spatio-temporal gait parameters of hospitalized older pationts: comparaison of fallers

and non-fallers

[Comments]

Thank you for the good research.  It seems that the paragraph of the thesis needs to be

corrected.  Overall, the length of paragraphs is short, hindering understanding of the content.

I think it will be a better thesis if it is modified according to the comments below.

[Introduction section]

1.  Learning to walk (line 29) ~ gati to predefinec standards (line 37): Please describe

in one paragraph.  And if the specific risk of falls in the elderly is presented, the need

for research will be further highlighted.

2.  These modifications, ~ gait parameters (line 38~40): This is the content that does not

connect with the preceding and following paragraphs.  Please modify it according to the

overall flow.

3.  These modifications, ~ gait parameters (line 38~40): This is the content that does not

connect with the preceding and following paragraphs.  Please modify it according to the

overall flow.

4.  GAITRite® as described for research purpose (line 53): The description of the

equipment used in the experiment is insufficient.  Specifically, it is not possible to

confirm the reliability and validity of GAITRite® , and it seems necessary to present the

contents of previous research using this equipment.

[Materials and Methods section]

5.  2.3.  Gait parameters (line 76~82): The variables that explain [Gait parameters] are

described in [2.4 Data collection].  It is correct to view the contents of the paragraph as

a description of the tas.  Therefore, the subheading needs to be corrected.

6.  2.5.  Data analysis: Please describe the statistical processing method for the same

variable in one paragraph.

7.  In this session, it is necessary to explain in detail how to measure gait using

GAITRite® .

[Results section]

8.  Table 2: Gait parameters and comparison with reference values: Why did you

check the statistical difference by calculating the difference amount without performing

[Reference value] and each group comparison?

9.  In all tables displayed in the thesis, presenting values by separating the right and

left sides with "/" is not easy to check the result value.  The table needs to be

reorganized.

[Discussion section]

10.  Fallers had a significantly higher history of stroke and osteoporosis than non-

fallers.  However, there was no significant difference in the gait parameters of the

two groups.  What do the authors think about why there is no significant difference

despite having a disease that can affect gait?  Please explain the above in [study

population characteristics] of the discussion.

Reviewer 2 Report

The manuscript "Spatio-temporal gait parameters of hospitalized older patients: comparaison of fallers and non-fallers" consists of 11 pages, including 5 tables and a list of 26 literature references. The authors examined spatio-temporal gait parameters of hospitalized older patients (with and without fall history) using the GAITRite mat.

Comments and suggestions:

Title

- There is a spelling mistake in title. The wording "comparaison" needs to be corrected in "comparison".

Abstract

- The authors state in abstract as follows: "The mean walking speed was 51.4 cm/s in non-fallers group, and 47.3 cm/s in fallers group (p = 0.539), indicating pathological walking in comparison with the general population of the same age." I think the authors can not aussume that every reader is firm with values of normal walking speed, so they should give a short remark.

Introduction

- The authors often use the term "the elderly". This term should be avoided it might be better to use "older adults" or "older people".

Materials and Methods

- The secondary end point was to compare spatio-temporal gait parameters between the study population and the general population."General population" is not specified in the manuscript. How is "general population" definded? Do authors mean adults or only older adults? Does this population also include adolescents or children? Where is the "general population" from? From authors' city/country?

- In section data analysis authors describe using Wilcoxon test. I think they mean Wilcoxon rank sum test (Mann-Whitney U test) for unpaired samples and not Wilcoxon signed rank test for paired samples. Please specify.

Results

- In table 1 in "rheumatology history, traumatic fractures" %-characters are double.

- Authors state in section "3.2 Gait parameters and comparison with reference values": "Walking speed, step and stride length were almost twice lower in the study population compared to the general population (about 100 cm/s), and the double support phase was twice longer in the study population." All these parameters of the reference population should be listed exactly. About 100 cm/s is an inaccurate description. In table 2 there are shown mean differences of gait parameters between general population and study population. I think raw data of both groups for these values could also be shown.

Discussion

- I appreciate authors' clear statement on study limitations (small sample size, decreased power).

- Line 227: I think " the main strength of our study were the use..." should be replaced into "the main strength of our study was the use.."

Round 2

Reviewer 2 Report

The authors made substantial improvements to the manuscript. I have one last minor point before publication. The spelling in the table submitted as supplementary file should be changed from France to English.
